# Foam Formation by Compression/Decompression Cycle of Soft Porous Media

**Phillip Johnson [1], Mauro Vaccaro [2], Victor Starov [1] and Anna Trybala [1,*]**

[1] Department of Chemical Engineering, Loughborough University, Loughborough LE11 3TU, UK; P.Johnson@lboro.ac.uk (P.J.); V.M.Starov@lboro.ac.uk (V.S.)
[2] Procter & Gamble, Temselaan 55, Grimbergen, B-1853 Brussels, Belgium; vaccaro.m@pg.com
[*] Correspondence: A.Trybala@lboro.ac.uk

**Abstract:** A theory of the amount of foam produced by compression/decompression cycles of a soft porous media is developed. The amount of foam produced was found to be dependent on both the amount of surfactant within the media and the minimum separation between the plates of the compression device. The latter is determined by the mechanical properties of the soft media. The theory also shows the importance of the decompression of the media as this is the mechanism of where the air penetrates into the soft porous material. The accumulated air is used during the compression stage for foam formation. The theoretically predicted values of foam mass are found to have good agreement with experimental observations, which validates the theory predictions. The theory also predicts independence of the foam produced in terms of the frequency of compression/decompression cycles, which agrees with our experimental observations.

**Keywords:** compression/decompression; soft porous media; foam formation; theory

## 1. Introduction

When a gas is injected into a liquid, bubbles are formed and a foam is created [1]. Foams are a dispersion of gas bubbles separated by thin liquid films [2–4]. In a pure liquid (water below), the foam is short-lived because the bubbles rapidly coalesce and foam collapses [5]. Therefore, stabilising agents are required to stabilise the generated foams [3,6–9]. Examples of stabilising agents are surfactants, particles, polymers, and their mixtures [10]. The most common of these are surfactants. Foams are used in many industries such as to make whipped cream, shaving cream, fire extinguishing materials, and in making medicine [1].

The foams that are focused on below are those produced by soft porous media, which links to the fabric and homecare industry. This particular soft compressive media has been recently investigated where the wettability of commercial dishwashing solutions were tested on soft porous media (sponges) [11] and the foamability of this solution on the porous media under compression [12].

Below the mechanism that drives the amount of foam is discussed and a mathematical model is developed. This model is compared to experimental results gathered for commercial dishwashing solution and sodium dodecyl sulphate (SDS).

## 2. Materials and Methods

The compression device previously developed [12] is presented in Figure 1. The device consists of two parallel plates controlled by pistons. The pistons are controlled by opening and closing valves controlling the airflow through the system. The pressure applied is controlled by manually turning dials for each piston where the pressure is monitored by a computer. The speed at which the plates

move is controlled by a regulator to vary the airflow to the piston where this speed is also monitored by the computer.

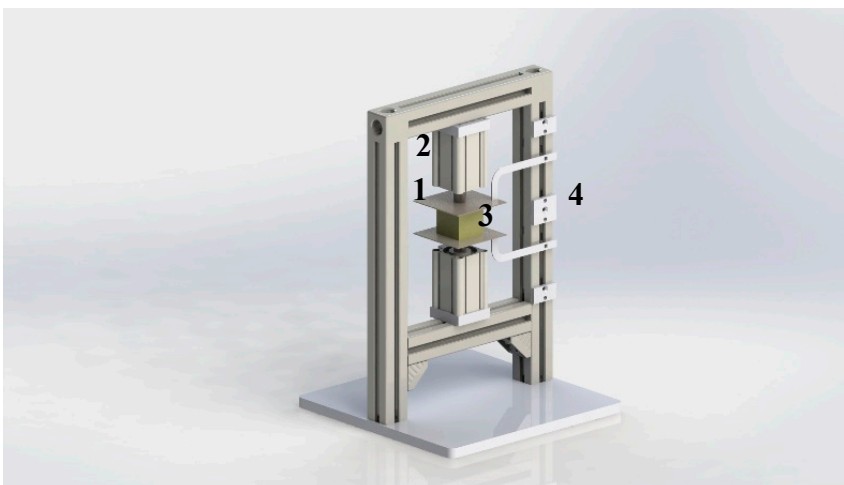

**Figure 1.** The compression device. 1—The metal plates. 2—Pistons. 3—Porous material. 4—Pressure controls [12].

The dishwasher sponge was used as a model of soft porous material and its properties are discussed below. The sponge is stated to be 100% polyester by the manufacturer. The properties of this sponge were determined using a scanning electron microscope (SEM). Porosity was calculated using MATLAB software by comparing the total area of the image to the area of the dark contrast pores. The average pore size was obtained using ImageJ software. An average pore diameter was calculated by measuring 45 pores from multiple SEM images of the media. The SEM images of the dishwasher sponge are presented in Figure 2. The pore size was found to be 0.2953 ± 0.0704 mm and the porosity was calculated to be 0.694 ± 0.013.

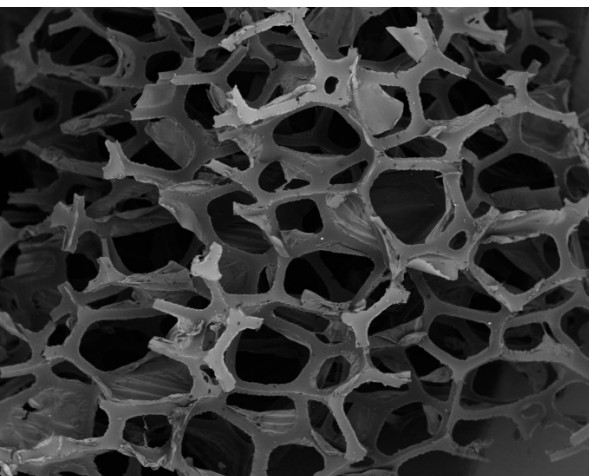

**Figure 2.** SEM image of the dishwasher sponge [12].

The total mass of the surfactant solutions for each concentration was 30 g, which is the total mass of liquid that a saturated piece of sponge, used in experiments, can hold without leakage. After adding the surfactant solution, the sponge was subjected to periodic cycles of compression/decompression. In this experiment, the foam is gathered after each compression into a weighted beaker immediately collecting the contents on the plates. The plates themselves have small holes cut into them to allow the liquid to drain but leave the foam behind. Due to this, the process of gathering only takes a

matter of seconds. In addition, since the amount of liquid remaining inside the porous media is lower than the mass observed without leakage, no liquid would drain during this gathering process. The compressions/decompression cycle will then be continued until no more foam is produced.

The surfactants used are either commercial dishwashing solution or Sodium Dodecyl Sulphate (SDS), from Fischer Scientific. The Commercial dishwashing solution is a mixture of anionic and non-ionic polymers where the exact composition is protected by the producer. SDS is an anionic surfactant where the critical micelle concentration (CMC) is 8.2 mMol/L or 0.24% (by mass) when dissolved in water. Concentrations investigated was 20% dishwashing solution to 80% water and 20 CMC (5.91g in 0.25 l of water).

## 3. Theory

It has been found that, in all experimental runs of the compression of sponges, no foam was produced during the first compression. After this, compression is then a decompression and then a further compression of this media led to the maximum amount of foam being produced during the second compression. The lack of foam produced during the first compression is due to the absence of air within the porous media. As the media is decompressed, the air flows into the media, which causes the formation of bubbles in the porous network. As the sponge reverts to its original shape, there is a continuous flow of air that creates an array of bubbles inside the sponge. During the second compression, the largest mass of foam is observed. The bubbles that are within the porous material after decompression are released in the succeeding compression.

*Modelling of Sponge Squeezing*

The pressure on the sponge is applied using the periodic movement of the upper plate (Figure 1). Displacement of the upper plate, F(t), is periodic (Figure 3). In Figure 3, a displacement of the upper plate varies from the top of the sponge of the thickness H to a minimum thickness, h, on the bottom of the sponge (Figure 3).

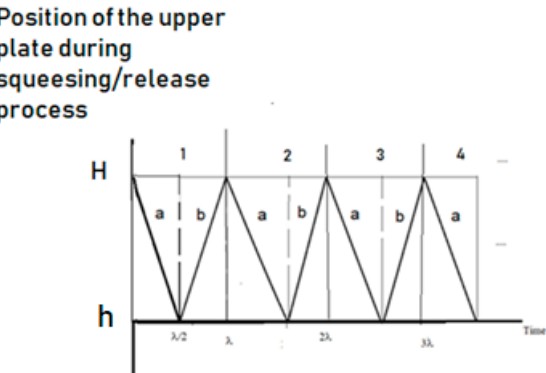

**Figure 3.** Graphical representation of the compression/decompression process of the foam production of the device shown in Figure 1 where a—compression stage and b—decompression stage.

The foam is produced only during the squeezing half period *a* (starting from the second stage as explained above). During the release part of the process, the air is moving from outside into the sponge and accumulates in the form of bubbles. During the first squeezing, the foam is not produced because there is no accumulated air inside the sponge.

Let the total volume of foam produced at the instant t is V(t). To produce this volume of foam, the following amount of surfactant, M(t), is consumed.

$$M(t) = M_l(t) + M_s(t) \tag{1}$$

where $M_l(t)$ is the total amount of surfactants inside the liquid part of the foam (Plateau borders, nods, lamellas).

$$M_l(t) = V(t) \cdot \varphi \cdot c \tag{2}$$

where $\varphi$ is the liquid volume fraction inside the foam and c is the concentration of surfactants in the liquid part of the foam. $M_s(t)$ is the amount of surfactant adsorbed on the total surface of bubbles inside the foam.

$$M_s(t) = (1 - \varphi) \cdot S \cdot \Gamma \tag{3}$$

where S is the total surface of bubbles inside the foam of volume $V(t)$ and $\Gamma$ is the adsorption of surfactants on the bubbles' surfaces. Substitution of Equations (2) and (3) into Equation (1) results in:

$$M(t) = V(t) \cdot \varphi \cdot c + (1 - \varphi) \cdot S \cdot \Gamma,$$

or

$$M(t) = V(t) \cdot [\varphi \cdot c + (1 - \varphi) \cdot S/V \cdot \Gamma].$$

However, $S/V \sim 1/R$, where R is the bubble radius. Hence, the latter equation can be rewritten as:

$$M(t) = V(t) \cdot [\varphi \cdot c + (1 - \varphi) \cdot \Gamma / R] \tag{4}$$

Considering the position of the upper plate, Equation (5) shows the position of the upper plate during the squeezing parts in Figure 3.

$$F_a = H - 2(H - h)(t - t_i)/\lambda \tag{5}$$

where H is the thickness of the sponge, $\lambda$ is the period, and time when the cycle i starts: $t_i = \lambda(i - 1)$; $i = 2, \ldots$

At the moment $t_i + \frac{1}{2\lambda}$, the upper plate reaches the bottom, that is, $F_a = h$ (Figure 3). The rate of surfactants' consumption to produce foam during cycle i, where $i = 2,3, \ldots$ at the moment t, $Q_{ia}(t)$, is proportional to (i) the mass of surfactants remaining in the sponge at the moment t in cycle i, $m_{ia}(t)$, and (ii) the applied pressure from the upper plate, which is proportional to $dF_a(t)/dt$. Hence, $Q_{ia}(t)$ can be expressed as:

$$Q_{ia}(t) = -\alpha \, dF_a(t)/dt \cdot m_{ia}(t) \tag{6}$$

where $\alpha$ is a proportionality coefficient (dimension 1/cm), which is determined by the properties of the sponge. It is possible to assume that the parameter $\alpha$ is inversely proportional to the final thickness of sponge h. The more the sponge is squeezed (i.e., smaller h), the more foam is produced. That is, we assume that $\alpha = const/h$, where the constant is a dimensionless value. It is shown below that this constant is equal to 1.

It is shown below that this parameter is equal to 1/h, where h is the minimum separation of the plates when fully compressed (Figure 3). h is determined by the compressibility of the porous media and the force applied.

The values of $M(t)$ and $Q(t)$ are related to the mass balance as follows.

$$\frac{dM_{ia}(t)}{dt} = Q_{ia}(t) \tag{7}$$

On the other hand, the produced foam consumed some surfactants, and the mass of surfactants decreased because of this. That is:

$$Q_{ia}(t) = -dm_{ia}(t)/d \tag{8}$$

Combining Equations (7) and (8), we arrive to the following differential equation for the unknown dependency of the mass of surfactants inside the sponge at cycle i at the moment t, that is, at the time between $\lambda(i - 1) < t < (i - 1 + 1/2)\,\lambda$ at the compressing half-cycle.

$$\frac{dm_{ia}(t)}{dt} = -\frac{2\alpha(H - h)}{\lambda} m_{ia}(t) \tag{9}$$

Equation (9) is the first-order differential equation and requires initial conditions at the moment, $t = \lambda(i - 1)$. These conditions are consequences of the conservation law for surfactants. That is, $m_{2a}(\lambda) = m_0$.

$$m_{ia}((i - 1 + 1/2)\lambda) = m_{i+1a}(i\lambda) \tag{10}$$

where $m_0$ is equal to the initial mass of surfactants deposited inside the sponge before the process starts.

In the following, we concentrate on the consideration only of the second cycle, Cycle 2, part a, time from $\lambda$ to $\lambda(1 + 1/2)$. The other cycles can be considered similarly, as shown below.

Let the initial mass of surfactants be m0. In this case, Equation (9) and boundary condition (10) take the following form:

$$\frac{dm_{ia}(t)}{dt} = -\frac{2\alpha(H - h)}{\lambda}{}_{a} m_{ia}(t) \tag{11}$$

$$m_{2a}(0) = m_0. \tag{12}$$

The solution of the latter equation using the initial condition (12) results in:

$$m_{2a}(t) = m_0 \exp(-2\alpha(H - h)(t - \lambda)/\lambda). \tag{13}$$

The mass of the remaining surfactants at the end of this half-cycle compression is:

$$m_{2a}(\lambda + \lambda/2) = m_0 \exp(-\alpha(H - h)) \tag{14}$$

The latter equation shows that the amount of surfactant left after the second cycle is (i) decreasing with the increase of the sponge thickness H and (ii) does not depend on the frequency of the squeezing/decompression process, $\lambda$.

Below measurements of the mass of foam produced during each cycle, $M_{ia}\left(t_i + \frac{\lambda}{2}\right)$ were provided. These amounts $M_{ia}\left(t_i + \frac{\lambda}{2}\right)$ should be predicted to compare with experimental measurements. From Equations (7) and (8): $\frac{dM_{2a}(t)}{dt} = -dm_{2a}(t)/dt$

Hence,

$$M_{2a}(t) = -m_{2a}(t) + \text{const} \tag{15}$$

The constant in the latter equation should be determined from the initial condition at the moment $t = \lambda$, when the mass of produced foam is equal to zero.

Hence, from Equation (15), we conclude const $= m_0$ and

$$M_{2a}(t) = m_0[1 - \exp(-2\alpha H(t - \lambda)/\lambda)] \tag{16}$$

According to Equation (16), the mass of foam collected after the second cycle, $M_2$, which can be measured, is:

$$M_2 = m_0[1 - \exp(-\alpha(H - h))] \tag{17}$$

The initial mass of the surfactants for the third cycle will be the following.

$$m_3 = m_0 \exp(-\alpha(H - h)).$$

Hence, the mass of the foam collected after the third cycle $M_3$ will be:

$$M_3 = m_0 \cdot \exp(-\alpha(H-h))(1-\exp(-\alpha(H-h))) \tag{18}$$

That is, during the third cycle, the amount of foam produced as compared with the second cycle, decreased by the factor $\exp(-\alpha(H-h))$.

Similarly, for any subsequent cycles, i = 3, the decrease by $\exp(-\alpha(H-h))$ continues:

$$M_i = m_0 \exp(-(i-2)\alpha A)(1-\exp(-\alpha(H-h))) \tag{19}$$

All dependencies $M_i(t)$ are similar but each next compression is multiplied by $\exp(-\alpha(H-h))$ as compared with the previous one (Figure 4).

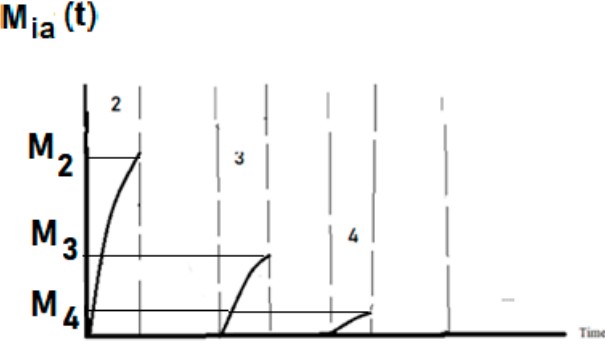

**Figure 4.** The predicted amounts of foam produced during each compression using Equation (19), starting from compression two as no foam is produced during the first compression.

## 4. Results and Discussion

The mass of foam produced after each compression is measured and then theoretically predicted values using Equations (17)–(19) are plotted alongside experimentally observed values of foam mass produced using commercial dishwashing solution (Figure 5). The mass of foam produced with each compression was observed. The compression forces were varied in different experimental runs. It was mentioned above that there is no foam produced during the first compression, which was expected and discussed previously. The maximum amount of foam produced was during the second compression and the foam mass exponentially decreased as the number of compressions continued to increase. The foam mass produced with each compression with 10 CMC SDS solution is also measured (Figure 6) and theoretical values are calculated and compared. In this investigation, the sponge used was the dishwasher sponge and the minimum separation of the plates when the compression occurs, h is shown by Table 1 for each force applied. The $\alpha$ value is then calculated as $\alpha = \frac{1}{h}$ for each value of h and is used in Equations (17)–(19) to calculate the theoretical plots for Figures 5 and 6. The table shows that the minimum thickness h is decreasing with applied pressure/force as expected. The theoretical curves in Figures 5 and 6 are plotted using Equation (19) where I was considered as a continuous parameter instead of as an integer.

**Table 1.** h value measured for each pressure applied and the calculated $\alpha$ value for each value of h.

| Pressure (bar) | h (cm) | $\alpha$ (1/cm) | 1/h (1/cm) |
|:---:|:---:|:---:|:---:|
| 0.56 | 0.4 | 2.5 | 2.5 |
| 1.55 | 0.3 | 3.33 | 3.33 |
| 2.5 | 0.2 | 5 | 5 |
| 3.5 | 0.1 | 10 | 10 |
| 4.5 | 0.1 | 10 | 10 |

Table 1 shows that the proportionality constant is equal to 1 and $\alpha = 1/h$.
The latter means that Equation (19) can be rewritten as:

$$M_i = m_0 \exp(-(i-2)A/h)(1 - \exp(-(H/h - 1))). \tag{20}$$

The latter equation shows a highly nonlinear dependency of the collected mass on the degree of compression h.

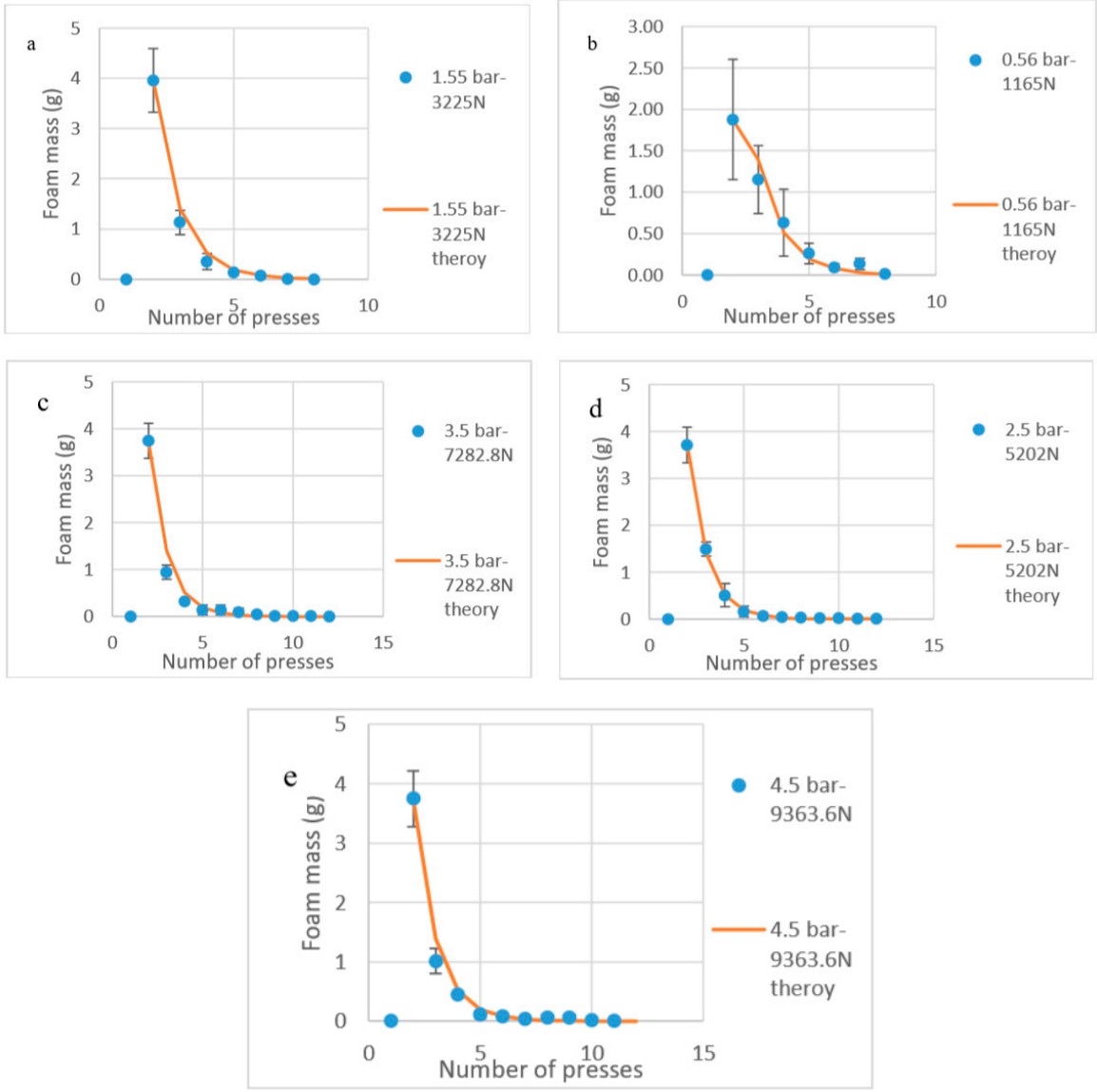

**Figure 5.** Foam mass for each compression of a dishwasher sponge saturated with 20% commercial dishwashing solution with different compression forces where (**a**) 0.56 bar = 1165 N, $\alpha$ = 2.5 (**b**) 1.55 bar = 3225 N, $\alpha$ = 1.333 (**c**) 2.5 bar = 5202 N, $\alpha$ = 5 (**d**) 3.5 bar = 7282.8 N, $\alpha$ = 10 (**e**) 4.5 bar = 9363.6 N. The Theoretical values are calculated using Equation (20).

In Figures 5 and 6, the first compression produces no foam. This was expected due to the fact that, in the saturated sponge, there is no air inside the completely saturated porous media. As discussed previously, it is actually during the decompression after the first squeeze that an array of bubbles is created inside the porous network. This is measured during the second compression. This process continues for later compressions, but with less foam produced due to the loss of surfactant from the porous media. This was demonstrated in the theory section where it was shown that, with continued

compressions, the amount of foam was dependent on the mass of the remaining surfactants in the media. The mass of foam was also dependent on $\alpha$, which is equal to 1/h where h is the minimum distance between the plates.

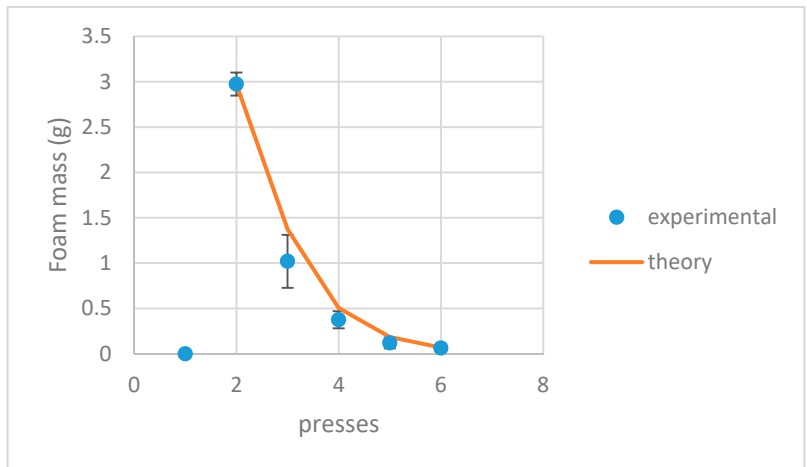

**Figure 6.** Foam mass for each compression of a dishwasher sponge saturated with 10 cmc SDS solution along with the theoretically predicted values using Equation (20). The pressure applied 2.5 bar = 5202 N.

The different velocity of the plates was also investigated to observe the effect the frequency of this compression/decompression cycle has on the amount of foam produced. In Figure 7, we showed an example of measurements at two different forces as an example.

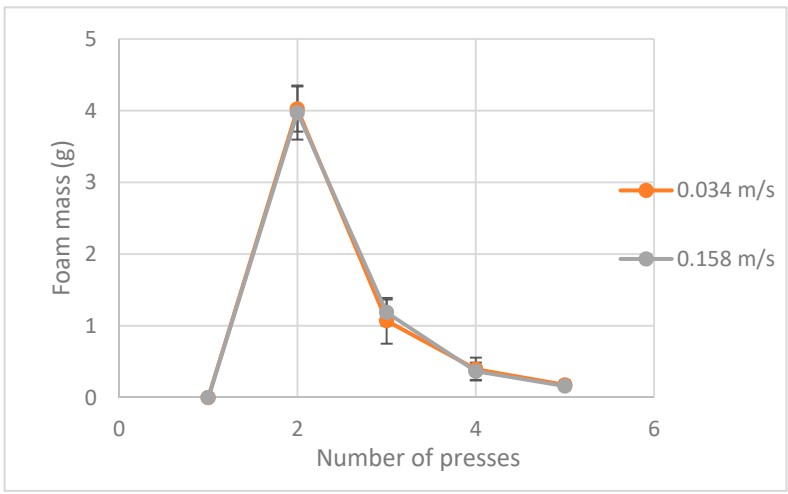

**Figure 7.** Foam mass for each compression of a dishwasher sponge saturated with 20% commercial dishwashing solution with different compression rates.

From Figure 7, it shows that there is no clear trend between the amount of produced foam and the rate of compression this agrees with as predicted theoretically. The affect rate of compression on foam quality, such as bubble size, is an area that needs further investigation.

Figures 5 and 6 show an agreement between the theoretically predicted values of foam mass after each compression and the experimentally observed ones. This demonstrates that the theory for foam mass produced by compression/decompression is in good agreement with experimental measurements for both the commercial dishwashing product and a basic surfactant (SDS). This means that, for the first time, the amount of foam produced by compression when using surfactant solutions can be calculated using the theory presented above. This could be of interest to companies that work with cleaning products as this industry uses a similar design for foam formation described above.

## 5. Conclusions

A new theory was developed for foam formation using squeezing of soft porous materials. The first compression of the media produces no foam as the media contains no air since all of the pores are saturated with solution. The amount of foam produced with each compression is dependent on the amount of surfactant remaining in the porous media and on the minimum separation between the plates. The minimum separation, h, is a consequence of the mechanical properties of the porous media and is used to calculate the only parameter used $\alpha = 1/h$.

There is good agreement between the theoretically calculated foam mass and the experimentally observed values for both commercial dishwashing solution and SDS.

This allows, for any concentration of surfactant, the ability to calculate the mass of foam produced when using compression of porous material as the foam producing method. For each compression and by summing up all the masses, the total foam mass that can be produced can also be predicted.

According to the theoretical prediction, the mass of foam produced during compression/decompression cycles does not depend on the frequency of the process, which is confirmed experimentally.

**Author Contributions:** Conceptualization, V.S. and A.T.; Methodology, P.J.; Validation, P.J. and A.T.; Formal Analysis, P.J.; Investigation, P.J.; Resources, A.T. and M.V.; Data Curation, P.J., Writing—Original Draft, V.S. and P.J.; Writing—Review and Editing, A.T., V.S., M.V. and P.J.; Visualization, P.J., V.S.; Supervision, A.T. and V.S.; Project Administration, A.T. All authors have read and agreed to the published version of the manuscript.

**Funding:** This research was funded by This research was supported by MULTIFLOW and CoWet EU grants, PASTA, MAP EVAPORATION European Spac and Proctor & Gamble, Brussels.

**Acknowledgments:** Proctor & Gamble, Brussels, and Map Evaporation Project, ESA, supported this research.

**Conflicts of Interest:** The authors declare no conflict of interest.

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
