# Peer review of "Foam Formation by Compression/Decompression Cycle of Soft Porous Media"

_colloids, doi:10.3390/colloids4030031_

Round 1

Reviewer 1 Report

This manuscript describes foam generation if a sponge, saturated with surfactant solution, undergoes a compression-decompression cycle. The authors present experimental data and a model for mass of produced foam as a function of cycle number.

Unfortunately I am not able to recommend publication of the manuscript for several reasons. My main criticism concerns the development of the model. Ultimately the authors aim to explain the roughly exponential decay in the amount of foam produced as a function of cycle iteration. Would this not simply follow from the assumption that the produced foam is proportional to the amount of liquid left in the sponge? Each squeeze reduces this, leading to exponential decay. I cannot see that the model, expanded in 19 equations, produces a theoretical understanding of the decay constant.

Further detailed criticism.

Would one not expect alpha to be inversely proportional to h? The more the sponge is squeezed (i.e. smaller h), the more foam is produced?

Eqn 4, S/V ~ 1/R, I agree with the scaling, but when using this relation further on a proportionality constant would be required.

There is also a lack of detail with regard the experimental procedure. How is the mass of the foam determined? The authors only write, “the foam is gathered” after each compression. How is this done, how long does it take? Is the sponge dripping in the meantime?

It is stated that alpha is determined by the properties of the sponge. In the conclusions it is referred to mechanical properties. I find this too vague.

The conclusions also state that “the mass of foam produced does not depend on the frequency of the process”.  I cannot see where this is backed up by data, figures 5 and 6 have no reference to time-scale of the compression cycles.

Reviewer 2 Report

The submitted manuscript entitled ‘Foam formation by compression/decompression cycle of soft porous media’ deals with the foaming process generated by the cyclic load of a polymeric sponge. The manuscript sounds and worth to publish, during ist review only a few minor issues arose as listed below.

- ‘Porosity was calculated using MATLAB software by comparing the total area of the image to the area of the dark contrast pores.’ – was it done on 2D images or 3D structures?

- Please add scalebar to fig 2.

- ‘The pore size was found to be 0.2953 ± 0.0704 mm…‘ – was the pore size really measured down to the tenth of microns?

- Fig 3 can be omitted since it is carrying very limited scientific information.

- The subfigs of fig 6 could be merged to spare space.

- The whole manuscript should be carefully proof read: there are a number of double spaces, space before stops and the fontsize is changing in lines 110, 131 and 135, etc.

Reviewer 3 Report

The manuscript presents a further development of the foamability of soft porous media using compression as presented in Ref. [12]. The focus is on the mechanism that determines the overall amount of foam quantity and its verification through experimental outcomes for a commercial dishwashing formulation and a model surfactant (SDS). The accent is on modelling of sponge squeezing.

The manuscript is of interest for the audience of MDPI Colloids and interfaces. It may be published in the present form after some minor corrections, namely

  1. The first paragraph in the Theory section (p.3, lines 74-82) has to be rephrased. There are too many typos and unclear statements.
  2. On p.6 the caption to Table 1 should not mention “force”, as the respective values are erased.
  3. On p. 9, lines 250-252 the paragraph is unclear and should be rephrased.

Round 2

Reviewer 1 Report

I am not able to recommend publication of this revised version of the manuscript.

The treatment of alpha is unclear to me. In table 1 it appears that alpha is defined as 1/h and then the authors also show the value of 1/h, which by definition has to agree with alpha? This is odd.

The benefit of Eqn 20 (eqn 19 in the current manuscript) appears to be that it is highly nonlinear, highly non-linear in what? I can not see that the experimental data supports this equation.

The caption for the new figure (7/8) contains several spelling mistakes.
